## Comment

behaviour/cognition/computational biology

formal language theory, expressive power, compactness, animal song, human language

**Authors for correspondence:**
Aniello De Santo
e-mail: aniello.desanto@stonybrook.edu
Jonathan Rawski
e-mail: jonathan.rawski@stonybrook.edu

# What can formal language theory do for animal cognition studies?

Aniello De Santo[1,2] and Jonathan Rawski[1,2]

[1]Department of Linguistics, Stony Brook University, and [2]Institute for Advanced Computational Science, Stony Brook University, Stony Brook, NY, USA

ADS, 0000-0001-6568-9919

There is a long tradition of studies using formal grammars to probe differences between human language and animal song, in terms of expressive power (*generative capacity*). In this sense, previous literature argues that the correct upper bound to the complexity of animal song is the regular class, and that supra-regular expressive power is unnecessary to capture patterns in animal communication systems.

Morita & Koda (henceforth also M&K; [1]) attempt to resurrect supra-regular analyses of animal pattern recognition. First, they state that claims about the regularity of animal song are *not supported by empirical evidence*, as supra-regular analyses of animal patterns are *possible*; second, they analyse gibbon data via probabilistic context-free grammars (PCFG; a notoriously supra-regular formalism), and invoke compactness of the analysis as a fundamental advantage of this approach.

Here, we discuss how the arguments raised by M&K are either (a) mathematically trivial or (b) mathematically ambiguous, resulting in misleading inferences about the cognitive questions at the core of this research programme. We do not address their probabilistic modelling results per se, as even the most sound statistical analyses are made uninterpretable by conceptual fallacies. We conclude with suggestions for the contribution of formal language theory (henceforth, FLT) to comparative cognition.

Let us start from their claim that arguments for the regularity of animal song are not empirically supported. The fact that supra-regular characterizations of regular patterns exist is a trivial observation. Given any regular language, it is always possible to write a context-free grammar that characterizes said language. These complexity classes are in a strict subset relation with respect to each other, so members of weaker classes are also members of more powerful classes [2].

One key finding in the paper—namely, that PCFGs do not improve fit to the gibbon data compared to linear models—falls out immediately from this: the additional hierarchical information intrinsically provided by the context-free analysis is simply unnecessary to describe the data.

In linguistics, FLT results are used to characterize the *minimally* expressive class of grammars able to generate the class of possible human languages. This is done to understand the fundamental properties of the language system (i.e. invariant with respect to representations and devices we might use to encode a pattern). Interest in contrasting human language and animal song patterns comes from the insight that such comparisons tell us something about similarities and differences in the underlying cognitive architectures. Focusing on more expressive characterizations instead of the minimally expressive ones (e.g. moving from regular to context-free) obfuscates these questions, by confusing what is possible with what is sufficient.

This point holds even if we accept the suggestion that arguments for the supra-regularity of human language are fallacious, since they rely on unrealistic assumptions about unbounded syntactic constructions (see [3,4] for empirical counterarguments to that suggestion, and [5,6] for conceptual ones). Moreover, no known animal song pattern seems to necessarily require supra-regular analysis, even with the hypothetical idealization allowed in the analyses of human language.

The authors seem to acknowledge the problem, as they do not make the existence of more powerful characterizations their central argument. Instead, they argue that the advantage of analyses adopting a more powerful formalism lies in their compactness. However, it is unclear how (or whether) the compactness of a characterization relates to minimally adequate computations for a specific pattern in meaningful ways [7].

Trade-offs between a formalism's expressive power and descriptive complexity—efficiency of describing languages—have been extensively studied in FLT (see [8, a.o.] for a review). For instance, context-free descriptions of regular languages can lead to exponential improvement in the size of the recognition machine [9,10]. However, this potential gain in succinctness comes as a byproduct of the additional structure needed by the recognition mechanisms (e.g. a more complex memory system, corresponding to adding a push-down store to a simpler finite automata). In other words, a smaller grammar pays the price of additional computational resources [11,12, a.o.]. Any compactness claim made without explicit assumptions about the underlying computational resources is thus uninteptable.

Since such resources are what formal studies of linguistic patterns aim to uncover, M&K's line of reasoning is not just unfounded, but pushes the scientific community towards misguided—and, dangerously, empirically unfalsifiable—conclusions. Mathematical models are only helpful to cognitive investigations if we remain aware of our ignorance about the constraints imposed by the underlying cognitive machinery, thus finding ways to gain insights into the computational resources needed by the process under study [13].

What can FLT contribute to comparative cognition then? In our view, the major insights come from learning studies. There are close links between formal language classes, the grammars that generate them, and their learnability. In particular, one can rigorously define necessary and sufficient conditions for successful learning of different complexity classes (biases towards specific data structures, amount of time and data etc.) [14]. These connections between available resources and possible (or impossible) learning outcomes allow us to circumvent the issues mentioned above about the opaqueness of the underlying cognitive architecture for humans and other animal species.

This is directly revealed by artificial grammar learning experiments, where subjects are tasked with generalizing from a small data sample to a productive grammar. Since the kinds of patterns subjects generalize over are implicitly constrained by the available computational resources, the string sets they arrive at are indicative of the capacity of the underlying cognitive machinery. If subjects of a particular species consistently generalize to patterns within a given complexity class, we may conclude that the species has ability to recognize string sets of at least this level of complexity, and vice versa [15].

This is not a novel suggestion, as these studies are extensively employed in probing both human and animal linguistic abilities [16]. What we advocate, in the spirit of M&K's paper, is closer collaboration between mathematical linguists and cognitive scientists. Once sound, theoretically informed hypotheses are laid down, comparative analyses may benefit from the more advanced techniques M&K argue for.

Data accessibility. This article has no additional data.
Competing interests. We declare we have no competing interests.
Funding. No funding has been received for this article.
Acknowledgements. The authors thank Thomas Graf and Jeffrey Heinz for insightful comments.

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
