## [Reviewer comments · Royal Society Open Science]

Review History

RSOS-191772.R0 (Original submission)

Review form: Reviewer 1

Is the manuscript scientifically sound in its present form?

Yes

Are the interpretations and conclusions justified by the results?

Yes

Is the language acceptable?

Yes

Do you have any ethical concerns with this paper?

No

Have you any concerns about statistical analyses in this paper?

No

Recommendation?

Accept as is

Comments to the Author(s)

This comment on Morita & Koda (M&K) is well written, and in my view a good contribution.

I believe there would be grounds for disagreement with the authors of this comment, but not of the kind that would warrant any request for changes, especially given the (short) nature of a comment of this kind. I believe they make good points that should generate debate and suggest the comment be published in its current form.

- in my view, the disagreement lies at what is necessary for saying that any one song or pattern recognition in a species *actually is* regular or super-regular.

- For the authors of this comment (and, naturally, many others), if something can be described regularly, one can or must assume it is regular, and any supra-regular analysis is unnecessary and unjustified. For Morita & Koda, a suprar-regular analysis of a system that can be analyzed as regular yields the possibility that the species being studied might actually have supra-regular "abilities", because we don't *actually* know what the species is doing.

- Perhaps M&K's suggestion is that FLS offers valuable descriptive tools for animal communication systems, but that there is not much one can readily apply to the biological study of the species that employ them.

- The authors of this comment make an interest comment on the claims of compactness made by M&K (the simpler the grammar, the more complex the architecture required for it to be computed). It is a good point, but since this increase in complexity is at the algorithmic level in might not be relevant for M&K's purposes.

I believe the (potential) disagreements between the authors can generate interesting debate. Those interested in the debate will perhaps be compelled to make more explicit ontological commitments when discussing the relationship between formal language theory and animal and human abilities.

I reiterate my recommendation to accept this comment in its current form.

Review form: Reviewer 2

Is the manuscript scientifically sound in its present form?

Yes

Are the interpretations and conclusions justified by the results?

Yes

Is the language acceptable?

Yes

Do you have any ethical concerns with this paper?

No

Have you any concerns about statistical analyses in this paper?

No

Recommendation?

Accept as is

Comments to the Author(s)

The authors of this commentary on the paper by Morita & Koda raise a valid and important issue. In their paper Morita & Koda argue that gibbon vocalizations can be described by using a supra-regular grammar. M&K present this as an alternative for the use of regular grammars that had so far been proven sufficient to describe the structure of animal vocalizations. They argue that such analyses of animal vocalizations have rarely been attempted but allow for a better comparison with human language.

The authors of the current commentary rightly point to an important scientific principle: one should refrain from invoking a more complex explanation of a phenomenon if a more simple one suffices. The gibbon vocalizations analyzed by Morita & Koda can be described by using a finite state (regular) grammar. Hence using a higher order supra-regular grammar is not a necessity. And, as the authors of the commentary point out: in the formal language theory any more powerful grammar can always deal with any feature of the weaker ones. The commentary also make clear that the results of the original paper provide no compelling argument to justify that using a supra-regular grammar should be preferred over a regular one.

So, I agree with the authors of the commentary. They point at an important and fundamental problem with the original article. They argue their case convincingly and this comment deserves to be published.

Decision letter (RSOS-191772.R0)

05-Nov-2019

Dear Mr De Santo:

It is a pleasure to accept your manuscript entitled "What Can Formal Language Theory Do for Animal Cognition Studies?" in its current form for publication in Royal Society Open Science. The comments of the reviewer(s) who reviewed your manuscript are included at the foot of this letter.

To ensure that the processing of your paper is as swift as practical, please email the editorial office an editable file version of your manuscript (Word or LaTeX are preferred).

In addition, please note the authors of the original paper are invited to submit a Reply to this Comment, and your Comment will be published alongside the Reply.

Kind regards,
Andrew Dunn

Senior Publishing Editor
 Royal Society Open Science Editorial Office
 Royal Society Open Science
 openscience@royalsociety.org

on behalf of Dr Claudia Wascher (Associate Editor) and Dr Kevin Padian (Subject Editor).

Associate Editor Dr Claudia Wascher Comments to Author:

Associate Editor: 1

Comments to the Author:

The authors provide a commentary to a recent paper by Morita & Koda (2019) published in Royal Society Open Science and the use of supra-regular analyses of animal vocalisations. In their commentary, the authors challenge the findings of the original study and argue for alternative explanations. The reviewers find the presented commentary well written and while potentially controversial, a valuable contribution to generate an interesting debate

Reviewer(s)' Comments to Author:

Reviewer: 1

Comments to the Author(s)

This comment on Morita & Koda (M&K) is well written, and in my view a good contribution.

I believe there would be grounds for disagreement with the authors of this comment, but not of the kind that would warrant any request for changes, especially given the (short) nature of a comment of this kind. I believe they make good points that should generate debate and suggest the comment be published in its current form.

- in my view, the disagreement lies at what is necessary for saying that any one song or pattern recognition in a species *actually is* regular or super-regular.

- For the authors of this comment (and, naturally, many others), if something can be described regularly, one can or must assume it is regular, and any supra-regular analysis is unnecessary and unjustified. For Morita & Koda, a supra-regular analysis of a system that can be analyzed as regular yields the possibility that the species being studied might actually have supra-regular "abilities", because we don't *actually* know what the species is doing.

- Perhaps M&K's suggestion is that FLS offers valuable descriptive tools for animal communication systems, but that there is not much one can readily apply to the biological study of the species that employ them.

- The authors of this comment make an interest comment on the claims of compactness made by M&K (the simpler the grammar, the more complex the architecture required for it to be computed). It is a good point, but since this increase in complexity is at the algorithmic level in might not be relevant for M&K's purposes.

I believe the (potential) disagreements between the authors can generate interesting debate. Those interested in the debate will perhaps be compelled to make more explicit ontological commitments when discussing the relationship between formal language theory and animal and human abilities.

I reiterate my recommendation to accept this comment in its current form.

Reviewer: 2

Comments to the Author(s)

The authors of this commentary on the paper by Morita & Koda raise a valid and important issue. In their paper Morita & Koda argue that gibbon vocalizations can be described by using a supra-regular grammar. M&K present this as an alternative for the use of regular grammars that had so far been proven sufficient to describe the structure of animal vocalizations. They argue that such analyses of animal vocalizations have rarely been attempted but allow for a better comparison with human language.

The authors of the current commentary rightly point to an important scientific principle: one should refrain from invoking a more complex explanation of a phenomenon if a more simple one suffices. The gibbon vocalizations analyzed by Morita & Koda can be described by using a finite state (regular) grammar. Hence using a higher order supra-regular grammar is not a necessity. And, as the authors of the commentary point out: in the formal language theory any more powerful grammar can always deal with any feature of the weaker ones. The commentary also make clear that the results of the original paper provide no compelling argument to justify that using a supra-regular grammar should be preferred over a regular one.

So, I agree with the authors of the commentary. They point at an important and fundamental problem with the original article. They argue their case convincingly and this comment deserves to be published.
